# Comparative Characterization of *Plasmodium falciparum* Hsp70-1 Relative to *E. coli* DnaK Reveals the Functional Specificity of the Parasite Chaperone

**DOI:** 10.3390/biom10060856

**Published:** 2020-06-04

**Authors:** Charity Mekgwa Lebepe, Pearl Rutendo Matambanadzo, Xolani Henry Makhoba, Ikechukwu Achilonu, Tawanda Zininga, Addmore Shonhai

**Affiliations:** 1Department of Biochemistry, School of Mathematical & Natural Sciences, University of Venda, Thohoyandou 0950, South Africa; charity.lebepe@gmail.com (C.M.L.); pearlrutendo@gmail.com (P.R.M.); tzininga@gmail.com (T.Z.); 2Department of Biochemistry, Genetics and Microbiology, University of Pretoria, Pretoria 0028, South Africa; xolani.makhoba@up.ac.za; 3Protein Structure-Function Research Unit, School of Molecular and Cell Biology, University of the Witwatersrand, Johannesburg 2050, South Africa; Ikechukwu.Achilonu@wits.ac.za; 4Department of Biochemistry, Stellenbosch University, Stellenbosch 7602, South Africa

**Keywords:** *Plasmodium falciparum*, Hsp70, co-expression, chaperone function, specificity

## Abstract

Hsp70 is a conserved molecular chaperone. How Hsp70 exhibits specialized functions across species remains to be understood. *Plasmodium falciparum* Hsp70-1 (PfHsp70-1) and *Escherichia coli* DnaK are cytosol localized molecular chaperones that are important for the survival of these two organisms. In the current study, we investigated comparative structure-function features of PfHsp70-1 relative to DnaK and a chimeric protein, KPf, constituted by the ATPase domain of DnaK and the substrate binding domain (SBD) of PfHsp70-1. Recombinant forms of the three Hsp70s exhibited similar secondary and tertiary structural folds. However, compared to DnaK, both KPf and PfHsp70-1 were more stable to heat stress and exhibited higher basal ATPase activity. In addition, PfHsp70-1 preferentially bound to asparagine rich peptide substrates, as opposed to DnaK. Recombinant *P. falciparum* adenosylmethionine decarboxylase (PfAdoMetDC) co-expressed in *E. coli* with either KPf or PfHsp70-1 was produced as a fully folded product. Co-expression of PfAdoMetDC with heterologous DnaK in *E. coli* did not promote folding of the former. However, a combination of supplementary GroEL plus DnaK improved folding of PfAdoMetDC. These findings demonstrated that the SBD of PfHsp70-1 regulates several functional features of the protein and that this molecular chaperone is tailored to facilitate folding of plasmodial proteins.

## 1. Introduction

Heat shock protein 70 (Hsp70) molecular chaperones are involved in protein folding. *Plasmodium falciparum* Hsp70-1 (PfHsp70-1) is a cytosol-localized molecular chaperone that is essential for survival of the malaria parasite [1,2]. PfHsp70-1 has been proposed as a prospective antimalarial drug target [2,3,4]. Furthermore, PfHsp70-1 is implicated in antimalarial drug resistance, making its inhibition using antimalarial drug combinations promising [5]. Although some compounds that inhibit PfHsp70-1 reportedly target the parasitic protein, exhibiting minimum effects on human Hsp70 [4], the unique structure-function features of this protein remain to be fully explored.

Structurally, Hsp70 consists of two functional domains: an N-terminal nucleotide-binding domain (NBD) and a C-terminal SBD [6]. The NBD of Hsp70 binds to ATP, hydrolyzing it to ADP [7]. On the other hand, the SBD binds to the peptide substrate. The SBD of Hsp70 is sub-divided into α and β-subdomains. The Hsp70 SBDβ makes direct contact with the peptide substrates while the SBDα serves as a lid enclosing the bound substrate [8]. Hsp70 exhibits high affinity for substrate in its ADP-bound state and releases it upon binding to ATP [6]. Therefore, nucleotide binding regulates substrate binding and release by the Hsp70 chaperone. Hsp70 possesses weak basal ATPase activity and hence relies on a co-chaperone, Hsp40, which stimulates its ATPase activity [9]. In addition, Hsp40 binds to mis-folded proteins first before transferring them to Hsp70 for refolding [10]. Thus, delivery of substrate to Hsp70 is concomitantly linked to ATP hydrolysis [11].

*Escherichia coli* Hsp70 (DnaK) structurally constitutes a canonical Hsp70, characterized by a conserved NBD connected to the SBD via a highly conserved linker motif [12]. Although DnaK is not essential for *E. coli* growth at intermediate temperatures, it is essential at high growth temperatures [13]. Both PfHsp70-1 and *E. coli* DnaK are cytosolic chaperones. PfHsp70-1 and a chimeric protein, KPf (made up of the ATPase domain of DnaK and the SBD of PfHsp70-1), were previously shown to protect *E. coli dnaK756* cells (harboring a functionally compromised DnaK) against heat stress [14]. In addition, PfHsp70-1 has been shown to provide cyto-protection to yeast cells endowed with a defective Hsp70 [15]. Altogether, this suggests that although Hsp70s are functionally specialized, they also exhibit functional overlaps across species. For example, it was previously reported that PfHsp70-1 possesses higher basal ATPase activity compared to its human and *E. coli* Hsp70 homologues [3]. The unique structure-function features of PfHsp70-1 in comparison to human Hsp70, have spurred interest to target it as part of anti-malarial drug design efforts [3,4,16].

*P. falciparum* Hsp40 (PF3D7_1437900) is a co-chaperone of PfHsp70-1 that co-localizes with it to the parasite cytosol [17]. PfHsp40 is regarded as a member of the so-called type I Hsp40s on account its structural resemblance to *E. coli* Hsp40 (DnaJ; [18]). Both DnaJ and PfHsp40 possess a highly conserved J domain that facilitates cross-talk with Hsp70 [18]. It has been shown that Hsp40 chimeric proteins with J domains of variable eukaryotic origin cooperated with DnaK to confer cyto-protection to *E. coli* cells that were void of endogenous DnaJ [19]. This suggests that the highly conserved J domain of Hsp40 is capable of modulating the function of Hsp70s of varied species origin. Although functional specificity of Hsp70s across species is generally regarded to be on account of their cooperation with several Hsp40 partners [20], we still do not understand how such conserved molecules are adapted to their function. It is further believed that of the two domains of Hsp70, it is the less conserved SBD that provides it with functional specificity [7].

It has been proposed that nearly 10% of *P. falciparum’s* proteome is characterized by prion-like repeats and that at least 30% of the proteome is characterized by glutamate/asparagine rich segments [21,22]. For this reason, it is thought that *P. falciparum* Hsp70s are adapted to fold and stabilize its mis-folding-prone proteome [23,24,25]. To this end, we previously demonstrated that a *P. falciparum* chaperone, PfHsp70-x, which is exported to the parasite-infected red blood cell [26], exhibits preference for asparagine rich peptides, further suggesting that *P. falciparum* Hsp70s are primed to bind mis-folded proteins of the parasite [25].

Both PfHsp70-1 and its chimeric product, KPf, have been shown to confer cyto-protection to *E. coli* cells harboring functionally compromised DnaK [27]. This suggests that PfHsp70-1 and KPf exhibit functional overlap with DnaK. On the other hand, PfHsp70-1 and KPf have both been employed to improve the quality and yield of recombinant proteins of plasmodial origin expressed in *E. coli* [24,28]. This suggests that although PfHsp70-1 and KPf exhibit functional overlap with DnaK, they could be tailored to facilitate folding of proteins of plasmodial origin. For these reasons, PfHsp70-1, DnaK and their chimeric protein, KPf, present a convenient model for studying the functional specificity of PfHsp70-1.

*P. falciparum* adenosylmethionine decarboxylase (PfAdoMetDC) is an essential protein involved in the biosynthesis of polyamines, making it a potential anti-malarial drug target [29]. Previously, we demonstrated that recombinant PfAdoMetDC co-expressed in *E. coli* with either KPf or PfHsp70-1 exhibited higher enzymatic activity than that co-expressed with supplementary *E. coli* DnaK [24].

GroEL and its cofactor, GroES, constitute a chaperonin of *E. coli* that is constituted of a cylindrical complex of two heptameric rings [30]. Thus, the GroEL/ES system constitutes a cage into which some mis-folded proteins are sequestered to facilitate their folding. It has been proposed that the GroEL/ES cage accommodates substrates of up to 60 kDa in size [31]. GroEL/ES and DnaK cooperate to facilitate folding of some proteins in *E. coli* [31]. For this reason, we investigated the effect of the three Hsp70s on the folding status of recombinant PfAdoMetDC expressed in *E. coli* BL21 Star^TM^ (DE3) cells. We further expressed each of the Hsp70 along with GroEL towards exploring their combined influence on PfAdoMetDC folding.

Our findings established that all the three Hsp70s exhibited comparable secondary and tertiary structures and they also shared some functional features. However, both PfHsp70-1 and KPf preferentially bound to peptide substrates that were enriched for asparagine residues while the presence of asparagine did not enhance the affinity of DnaK for these peptides. In addition, both PfHsp70-1 and KPf were marginally more stable to heat stress than DnaK. Our findings highlight the importance of the SBD of Hsp70 in stabilizing the conformation of this chaperone and its role in defining the functional specificity of the molecular chaperone. In addition, PfAdoMetDC co-expressed in *E. coli* with PfHsp70-1 and KPf exhibited similar biophysical features and was better folded than the protein co-produced with supplementary *E. coli* DnaK. This further demonstrates that the SBD of PfHsp70-1 could be structurally tailored to fold proteins of plasmodial origin.

## 2. Materials and Methods

### 2.1. Materials

Unless otherwise stated the reagents used in this study were procured from Sigma-Aldrich (Sigma Aldrich; Darmstadt, Germany) or ThermoScientific (Waltham, MA, USA). The following antibodies were used: rabbit raised polyclonal anti-PfHsp70-1 [32], mouse raised monoclonal anti-DnaK [27], mouse raised monoclonal anti-Strep-tag II [24] (Novagen, Madison, WI, USA), goat raised anti-rabbit horseradish peroxidase (HRP) conjugated (Sigma-Aldrich), goat raised anti-mouse HRP conjugated (Sigma-Aldrich; Darmstadt Germany), monoclonal α-His-HRP conjugated (ThermoScientific).

### 2.2. Comparative Analysis of Amino Acid Composition of AdoMetDC/ODC and Three-Dimensional Modelling of Hsp70s

Comparative amino acid composition was conducted for AdometDC/ODC from *P. falciparum* (PlasmDB accession number: PF3D7_1033100), *H. sapiens* (NCBI accession number: NP_001274118) and *E. coli* (NCBI accession number: WP_137659653.1). In addition, three dimensional models of the chaperones PfHsp70-1 (PlasmoDB accession number: PF3D7_0818900), DnaK and KPf were generated on Pyre2 using the template (2KHO) [12] and visualized using Chimera as previously described [16]. The generated models were subjected to structural comparison using the matchmaker tool on Chimera.

### 2.3. Expression and Purification of Recombinant Molecular Chaperones

Previously described plasmid constructs: pQE30/PfHsp70-1 (PlasmoDB accession number: PF3D7_0818900) encoding for PfHsp70-1 [32]; pQE30/KPf encoding for KPf [24]; pQE30/DnaK encoding for *E. coli* DnaK [27] were used for the expression of recombinant PfHsp70-1, KPf and DnaK proteins, respectively. In addition, a codon harmonized form of the PfHsp40 gene (PlasmoDB accession number: PF3D7_1437900) was cloned into pQE30 (Qiagen, Frederick, MD, USA) using *Bam*HI and *Hind*III restriction. The DNA segment encoding PfHsp40 was produced by GenScript (Piscataway, NJ, USA) and the integrity of the resultant pQE30/PfHsp40 was confirmed by agarose gel electrophoresis and DNA sequencing. The recombinant proteins were expressed in *E. coli* XL1 Blue cells in frame with N-terminally attached polyhistidine tags, which facilitated purification using affinity chromatography as previously described [33]. DnaK, PfHsp70-1 and KPf were successfully purified using sepharose nickel affinity chromatography under native conditions. PfHsp40 was purified as previously described [4]. The production of the (His)_6_-tagged recombinant proteins was confirmed by Western analysis using mouse monoclonal α-His-horseradish peroxidase conjugated antibodies (Sigma Aldrich). Imaging of the protein bands on the Western blot was conducted using the ECL kit (ThermoScientific) as per the manufacturer’s instructions. Images were captured using ChemiDoc Imaging System (Bio-Rad, Hercules, CA, USA).

### 2.4. Co-expression of PfAdoMetDC with Supplementary Molecular Chaperones

Next, we investigated the effect of KPf, PfHsp70-1 and DnaK on recombinant PfAdoMetDC folding upon expression in *E. coli*. This was conducted by co-expressing PfAdoMetDC in *E. coli* BL21 StarTM (DE3) cells with the following supplementary chaperone combinations: DnaJ-DnaK/PfHsp70-1/KPf plus/minus supplementary GroEL. To facilitate co-expression of PfAdoMetDC with DnaK + DnaJ, the pBB535 (encoding DnaK + DnaJ; kindly donated by Dr Bernd Bukau (Heidelberg University, Germany) and plasmid pASKIBA/PfAdoMetDC (supplied by Dr. Lyn-Marie Birkholtz, Pretoria University, South Africa) were used to co-transform *E. coli* BL21 Star (DE3) cells. As previously described, the DnaK encoding segment was substituted by DNA segments encoding for either KPf or PfHsp70-1 cloned onto the pBB535 [24]. Similarly, pBB542 (encoding for DnaK/PfHsp70/KPf + DnaJ + GroEL) was also used along with pASKIBA/PfAdoMetDC as previously described [24]. Thus, in summary, chemically competent *E. coli* BL21 StarTM (DE3) cells were co-transformed with pASKIBA/PfAdoMetDC along with either: pBB535 plasmid construct encoding for: PfHsp70-1 + DnaJ/KPf + DnaJ/DnaK + DnaJ or the plasmid pBB542 encoding for: PfHsp70-1 + DnaJ + GroEL/KPf + DnaJ + GroEL/DnaK + DnaJ + GroEL, respectively as previously described [24]. Briefly, the transformed *E. coli* BL21 Star (DE3) cells were cultured in LB broth supplemented with 100 μg/mL ampicillin (Sigma Aldrich) to select for pASKIBA3 and 50 μg/mL spectinomycin (Sigma Aldrich), to select for pBB535 and pBB542, respectively. As control, PfAdoMetDC expressed alone (without supplementary chaperones), was expressed in *E. coli* BL21 StarTM (DE3) cells as previously described [33]. The expression of chaperones was initiated by addition of 1 mM IPTG and at OD600 = 0.2 while the expression of PfAdoMetDC was initiated by addition of 2 ng/mL anhydroxytetracycline (AHT) (IBA GmbH, Göttingen, Germany) at OD_600_ = 0.7. Cells were incubated for 24 h (37 °C, 250 rpm) from the time of induction with IPTG. The cells were harvested by centrifugation (5000 *g*, 20 min, 4 °C) and the pellet was resuspended in lysis buffer (100 mM Tris pH 7.5; 300 mM NaCl; 10 mM Imidazole, containing 1 mM aminoethyl benzenesulfonyl fluoride hydrochloride (AEBSF) and 1 mg/mL of lysozyme). The recombinant PfAdoMetDC protein was purified using the Strep-Tactin Sepharose column (Invitrogen, Waltham, CA, USA) as described previously [24,34]. Purified PfAdoMetDC protein was quantified using the Bradford assay. Protein expression, solubility and purification were confirmed using SDS-PAGE analysis. Western blot analysis was used to verify the identity of PfAdoMetDC using monoclonal α-Strep-tag II antibodies (Novagen; Madison, WI, USA). Following Strep-Tactin Sepharose affinity column purification, PfAdoMetDC was analyzed using an Äkta explorer system (Amersham Pharmacia Biotech; Buckinghamshire, UK) fast protein liquid chromatography (FPLC), to determine migration profiles of the protein preparations that were co-expressed with the various chaperones sets. The migration pattern of PfAdoMetDC as determined by Size exclusion chromatography (SEC) was used to infer the compact/relaxed state of its fold. Retention times of the respective protein preparation were determined by SEC at 22 °C in running buffer (50 mM Tris-HCl, pH 7.5, 1 mM DTT, 500 mM NaCl and 0.02% *w*/*v* NaN_3_) using Bio-Select SEC 250-5 column (BioRad) with a fractionation range of 1000–250,000 Da. The column was briefly equilibrated with the running buffer and calibrated using BioRad gel filtration standards (thyroglobulin [670 kDa], bovine gamma globulin [158 kDa], chicken ovalbumin [44 kDa], equine myoglobin [17 kDa] and vitamin B12 [1.35 kDa]), and the standard covered MW range of 1350–670,000 Da. The elution profiles of the protein preparations were determined as peaks obtained at 280 nm using a Spectra Series UV100 spectrophotometer (Thermo Fischer Scientific).

### 2.5. Analysis of the Secondary Structures of the Recombinant Proteins

The secondary structure of the PfHsp70-1, DnaK and KPf protein was investigated using a far-UV circular dichroism (CD) (JASCO, Madrid, Spain) spectrometer as previously described [35,36]. Briefly, 0.2 μM of recombinant protein (DnaK, PfHsp70-1/KPf) in buffer PBS (137 mM NaCl, 2.7 mM KCL, 10 mM Na2HPO4, 2 mM KH2PO4, pH 7.5) was analyzed and spectral readings monitored at 190–240 nm. Spectra was averaged for 15 scans after baseline correction (buffer without recombinant protein). The spectra were deconvoluted to α-helix, β-sheet, β-turn and unordered regions, using the Dichroweb server, (http://dichroweb.cryst.bbk.ac.uk; [37]). The effect of heat stress on the secondary structure of the Hsp70s was investigated by monitoring the spectral changes of each protein at various temperatures (19–95 °C). The folded states of each Hsp70 protein at any given temperature were determined using spectral readings obtained at 222 nm as previously described [36]. In a separate experiment, the analysis of PfAdoMetDC protein co-expressed with the various chaperone sets was conducted by monitoring the CD spectrum at 190–240 nm at 22 °C. The assay was repeated in the presence of PfAdoMetDC substrate, S-adenosylmethionine hydrochloride (SAM), as previously described with minor modifications [38]. Briefly, PfAdoMetDC protein was preincubated in PBS supplemented with 100 μM SAM for 30 min prior to analysis at 25 °C. The resultant spectra were monitored and averaged following baseline correction (100 μM SAM using buffer void of the protein).

### 2.6. Fluorescence-Based Analysis of the Tertiary Structural Organization of the Recombinant Proteins

The tertiary structural conformations of DnaK, PfHsp70-1 and KPf proteins were assessed using tryptophan fluorescence as previously described [35]. The recombinant proteins (0.45 μM) were incubated in buffer HKKM (25 mM HEPES-KOH pH 7.5, 100 mM KCl, 10 mM MgOAc) for 20 min at 20 °C. Fluorescence spectra were recorded between 300 and 400 nm after initial excitation at 295 nm using a JASCO FP-6300 spectrofluorometer (JASCO, Tokyo, Japan). In addition, the effect of nucleotides on the conformation of each Hsp70 was investigated by repeating the assay in the absence or presence of 5 mM ATP/ADP. Similarly, biophysical characterization of PfAdoMetDC was conducted by monitoring both intrinsic (tryptophan and /tyrosine) and extrinsic (1-anilinonapthelene-8-sulfonate, ANS; Sigma Aldrich) emission spectra as previously described [39]. Since PfAdoMetDC possesses a single tryptophan residue and 33 tyrosine residues, the intrinsic PfAdoMetDC emission spectra were monitored at 300–400 nm. Furthermore, the extrinsic emission spectra generated upon ANS binding were conducted after an initial excitation at 370 nm of 2 μM PfAdoMetDC suspended in buffer HKKM supplemented with 100 μM ANS and monitoring the emission spectra at 400–600 nm. The assays were each repeated three times using independent batches of the respective protein.

### 2.7. Evaluation of ATPase Activities of DnaK, PfHsp70-1 and KPf

The basal ATPase activities of DnaK, PfHsp70-1 and KPf were evaluated based on the amount of released inorganic phosphate (pi) upon ATP hydrolysis as previously described [36,40]. Hsp70 proteins at final concentration of 0.4 μM were each incubated for 5 min in buffer HKMD (10 mM HEPES-KOH pH 7.5, 100 mM KCl, 2 mM MgCl2 and 0.5 mM DTT). The reaction was initiated by the addition of ATP at various concentrations (0–5 mM) and samples were analyzed every 30 min for 4 h. In order to determine the stimulatory effect of Hsp40 on the ATPase activity of Hsp70, the experiment was repeated in the presence of PfHsp40 (0.2 μM) as previously described [17].

### 2.8. Determination of the Nucleotide Binding Affinities of PfHsp70-1, DnaK and KPf

This assay was conducted at room temperature (25 °C) using a BioNavis Multi-parametric surface plasmon resonance (MP-SPR; BioNavis; Tampere, Finland) spectroscopy system. PfHsp70-1/DnaK/KPf (as ligands) were each immobilized through amine coupling on the functionalized 3D carboxymethyl dextran sensors (CMD 3D 500L; BioNavis). As analytes, aliquots of ATP/ADP were prepared at final concentration of 0, 1.25, 2.50, 5, 10 and 20 μM and were injected at 50 μL/min in each channel in series. Association was allowed for 2 min and dissociation was monitored for 8 min. Steady state equilibrium constant data were processed and analyzed using TraceDrawer software version 1.8 (Ridgeview Instruments; Uppsala, Sweden).

### 2.9. Investigation of Self-Association of Hsp70 Proteins

Self-association of the recombinant proteins was determined using a MP-SPR (BioNavis) as previously described [11,32]. As ligands, the recombinant Hsp70 proteins were immobilized on the CMD3D 500L chip. The analyte (respective Hsp70 protein) was prepared at various concentrations of 0, 125, 250, 500, 1000 and 2000 nM, respectively and injected at 50 μL/sec onto each channel with immobilized ligands. A reference channel without immobilized protein served as the control for non-specific binding and changes in refractive index. The analysis was conducted either in the absence or presence of 5 mM ATP/ADP. Association was allowed for 2 min, while dissociation was monitored for 8 min. Data analysis was conducted after double referencing by subtraction of both the baseline RU (buffer with ATP/ADP plus BSA as control protein) and RU from the reference channel. Kinetics steady-state equilibrium constant data were processed after concatenating the responses of all five analyte concentrations by global fitting using TraceDrawer software version 1.8 (Ridgeview Instruments).

### 2.10. Investigation of Interaction of PfHsp40 with DnaK, PfHsp70-1 and KPf

The direct interaction of each Hsp70 with PfHsp40 was analyzed using a MP-SPR (BioNavis), as previously described. As ligands, recombinant DnaK, PfHsp70-1 and KPf were immobilized at concentrations of 1.0 μg/mL per each immobilization surface. The analyte (PfHsp40) was prepared at varied concentrations of 0, 125, 250, 500, 1000 and 2000 nM and injected at 50 µL/sec onto the surface in series. The analysis was conducted both in the absence and presence of 5 mM of ATP/ADP. The ligand and analyte pairs were swapped, and the data generated were analyzed after global fitting. Association was allowed for 2 min, and dissociation was monitored for 8 min. Data analysis was conducted using TraceDrawer software version 1.8 (Ridgeview Instruments).

### 2.11. Interaction of DnaK, PfHsp70-1 and KPf with Model Peptide Substrates

The respective affinity of each Hsp70 for model peptide substrate was investigated by MP-SPR. The Hsp70 ligands were immobilized as previously described [36]. As analytes, the following model Hsp70 peptides were used: NRLLTG; NRNNTG; ALLLMYRR; ANNNMYRR; GFRVVLMYRF; and GFRNNNMYRF [25]. The analytes were injected at varying concentrations (0, 125 nM, 250 nM, 500 nM, 1000 nM and 2000 nM) over the immobilized ligands (Hsp70s). The analytes were injected at a flow rate of 100 µL/min and association and dissociation were allowed to occur for 10 min. The assay was conducted both in the absence and presence of 5 mM ADP/ATP. Analysis of association and dissociation data was conducted as previously described using TraceDrawer software version 1.8 (Ridgeview Instruments).

## 3. Results

### 3.1. Secondary and Tertiary Structural Analysis of DnaK, KPf and PfHsp70-1

The secondary and tertiary structural prediction based on the three-dimensional models of the Hsp70s show that the nucleotide binding domain is generally structurally conserved. However, the SBD show marked spatial orientations in space between the *E. coli* and the *P. falciparum* (Appendix A) proteins. Notably, compared to that of DnaK, the PfHsp70-1 SBD assumes a perpendicular (90° angle) orientation relative to the NBD. Interestingly, the chimeric protein KPf shows a similar orientation to DnaK in the NBD, while the spatial orientation of its SBD exhibited marginal variation from that of DnaK. The structural comparison between KPf and PfHsp70-1 is interesting, as both proteins share a similar SBD. Notably, the spatial orientations of their SBDs in space were distinct. The overall findings from these analyses suggest that the three-dimensional fold of Hsp70 is regulated largely by the NBD–SBD interface rather than by only the independent effects of the NBD or the SBD of the protein.

As a follow up, we conducted comparative secondary structure analysis of purified recombinant forms of DnaK, PfHsp70-1 and the chimeric KPf (Appendix A) using CD spectroscopy. The far-UV CD spectra exhibited negative troughs at 208–210 nm and 220–225 nm, representing the predominantly α-helical composition for all the three proteins (Figure 1a). A positive peak was also observed at 190 nm, representing the β-pleated sheets of the protein, as previously reported for PfHsp70-1 [35,41]. The secondary structure of each protein was assessed upon exposure to variable temperature conditions (19–95 °C) (Figure 1b). PfHsp70-1 and KPf maintained at least 50%-fold at temperatures up to 60 °C. However, DnaK rapidly unfolded at temperatures above 45 °C and completely lost its fold at approximately 68 °C. On the other hand, KPf and PfHsp70-1 became completely unfolded at around 90 °C. Our findings suggest that KPf and PfHsp70-1 were nearly equally resilient to heat stress at temperatures below 60 °C, and this is in agreement with a previous study [41], which reported that PfHsp70-1 is stable at temperatures above 50 °C. Notably, both KPf and PfHsp70-1 displayed a unique unfolding pattern, characterized by two steps for which the second phases occurred at temperatures 55–90 °C for KPf; and 75–90 °C for PfHsp70-1, respectively.

The tertiary structural conformations of DnaK, KPf and PfHsp70-1 were determined using tryptophan fluorescence either in the absence or presence 5 mM ATP/ADP. PfHsp70-1 has three tryptophan residues (W32, W101 and W593); KPf possesses two tryptophan residues (W102 and W578). On the other hand, DnaK possesses a single, W291 residue. Due to the varying tryptophan residue composition of the proteins, we focused on comparing the emission peaks rather than fluorescence quantum yields of the proteins (Figure 1c). DnaK gave an emission peak at 345 nm, and a marginal blue shift was observed in the presence of nucleotides (5 mM ATP/ADP) (Figure 1c). Similarly, PfHsp70-1 displayed an emission peak at 333 nm and exhibited a marginal blue shift in the presence of ATP/ADP (Figure 1c). On the other hand, KPf had an emission peak at 340 nm, and generated a significant blue shift in the presence of nucleotides (emission peaks of 335 nm in the presence of ADP; and 330 nm in the presence of ATP; Figure 1c). These data suggest that KPf exhibited a tertiary conformation that slightly varied to that of its parental forms. Furthermore, ATP had the most marked effect on the tertiary conformation of KPf than it had on DnaK and PfHsp70.

### 3.2. PfHsp40 Stimulates the ATPase Activities of All Three Hsp70s

The capability of DnaK, PfHsp70-1 and KPf to hydrolyze ATP independently was determined using a calorimetric assay (Figure 1d). The basal ATPase activity of DnaK was the lowest, while that of PfHsp70-1 was higher (*p* = 0.005) than that of DnaK. On the other hand, KPf registered the highest basal ATPase activity (Figure 1d; Appendix A). Next, we sought to explore the stimulatory effect of PfHsp40 on the ATPase activities of all the three Hsp70s. First, we validated that the PfHsp40 protein preparation had no detectable independent basal ATPase activity as expected. As a bona fide co-chaperone of PfHsp70-1, PfHsp40 modulated the ATPase activity of PfHsp70-1, registering more than ten-fold reduction in *Km* value compared to the basal ATPase activity of the chaperone (Appendix A in [17]). The highest ATPase activity was registered by KPf in the presence of PfHsp40 (Figure 1d, *p* < 0.01). On the other hand, the ATPase activity of DnaK was only marginally enhanced in the presence of PfHsp40. The chimeric protein, KPf, exhibited the highest ATPase activity (both basal and PfHsp40 stimulated). While Hsp40 primarily binds to the NBD domain to stimulate ATP hydrolysis by Hsp70, it is also known to make contact with the C-terminus of Hsp70 [42]. It is thus possible that the contact that PfHsp40 makes with the C-terminus of the SBD of PfHsp70-1 (also present in KPf) is unique compared to that of DnaK. This may account for the comparatively higher ATPase activities that both KPf and PfHsp70-1 register in the presence of PfHsp40 (*p* < 0.01). Whereas both PfHsp70-1 and KPf possess C-terminal EEVD residues known to make direct contact with Hsp40 [43], the equivalent C-terminal segment in DnaK is represented by residues, EEVKDKK.

### 3.3. KPf Exhibits Higher Affinity for ATP than either DnaK or PfHsp70-1

The relative nucleotide binding affinities of KPf, PfHsp70-1 and DnaK, were determined (Figure 1e). KPf exhibited a KD value which was approximately one order of magnitude higher than that of PfHsp70-1 and at least 200-fold higher than that of DnaK (Appendix A; *p* = 0.01). Since KPf and DnaK share the same NBD, the expectation would be that these two proteins exhibit comparable affinity for ATP. However, it is known that ATP binding at the NBD allosterically modulates Hsp70 to assume a closed conformation in which the C-terminal segment, in particular, the lid section, comes into contact with the NBD and this contributes towards ATP hydrolysis [44,45]. For this reason, the NBD cooperates with the SBD to influence both ATP binding and hydrolysis. This may explain why KPf exhibits much higher affinity for ATP than DnaK, despite the two proteins sharing the same NBD. Such a scenario highlights the role of the SBD–NBD interface in regulating ATP binding. The high affinity that KPf has for ATP could partly account for the marked blue shift in the tryptophan fluorescence signal that KPf displayed in the presence of ATP (Figure 1e).

### 3.4. Comparative Self-Association Capabilities

The self-association of Hsp70 is thought to be mediated by both the interdomain linker and the SBD [46,47]. For this reason, using SPR analysis, we investigated the capability of KPf to self-associate relative to its parental forms, DnaK and PfHsp70-1. The immobilized protein preparation served as a ligand while the protein in solution was used as the analyte. Our findings established that all the three proteins were capable of self-association (Table 1; *p* = 0.05). In addition, the oligomerization of the Hsp70s occurred in the absence of nucleotide and in the presence of 5 mM ADP/ATP (Table 1). However, the oligomerization of both PfHsp70-1 and KPf occurred with higher affinity in the presence of ATP than in the ADP or absence of nucleotides. On the other hand, in the presence of ATP, the affinity for DnaK self-association was one magnitude lower than that of either KPf or PfHsp70-1 (Table 1, *p* < 0.05). Overall, our findings suggest that KPf, DnaK and Hsp70-1 are all capable of forming higher order oligomers and that ATP enhances this process. Independent studies previously reported that DnaK forms higher order oligomers [48,49]. It is interesting to note that the fusion protein, KPf, retained this important functional regulatory feature of Hsp70.

### 3.5. All Three Hsp70s Directly Interacted with PfHsp40

PfHsp40 is a type I (structurally resembles *E. coli* DnaJ) co-chaperone of PfHsp70-1 [17]. In light of its high sequence homology to DnaJ and presence of the highly conserved J domain, PfHsp40 represents a typical Hsp40 whose propensity to interact with Hsp70s from various species such as plasmodial and human Hsp70 has been demonstrated in vitro [17]. In addition, in the current study we demonstrated that PfHsp40 stimulated the ATPase activities of DnaK, KPf and PfHsp70-1, further confirming its functional versatility (Figure 1d). We further sought to establish the interaction kinetics of PfHsp40 with DnaK, KPf, and PfHsp70-1, respectively. Using SPR analysis, we established that PfHsp40 directly binds to DnaK, PfHsp70-1 and KPf, respectively (Table 2; *p* = 0.05). Furthermore, the interaction was enhanced in the presence of ATP, in line with a previous independent study that demonstrated that Hsp70 association with Hsp40 is favoured in the ATP-bound state of Hsp70 [50].

### 3.6. PfHsp70-1 Preferentially Bound to Asparagine-Enriched Peptide Substrates

Approximately 10% of the malaria parasite proteome is thought to be characterized by prion-like repeats, and more than 30% of the proteome is characterized by glutamate/asparagine repeat segments [21,22]. Since PfHsp70-1 and KPf both possess an SBD of plasmodial origin and have previously been shown to support expression of recombinant plasmodial proteins in *E. coli* [24,28], we explored the comparative model Hsp70 peptide binding activities of KPf and PfHsp70-1 relative to DnaK. Thus, the binding kinetics of DnaK, KPf and PfHsp70-1 for a battery of synthetic peptide substrates were determined using SPR analysis (Figure 2; Appendix A). In this study, we employed canonical Hsp70 peptide substrates, (NRLLTG, ALLLMYRR, GFRVVLMYRF) [6,25,51,52] and repeated the assay using peptides that were modified by substituting the middle residues with asparagine residues (NRNNTG, ANNNMYRR, GFRNNNMYRF; [25]). We explored the interaction of each Hsp70 with each substrate either in the apo state or in the presence of ATP/ADP (Figure 2; Appendix A). ATP is known to suppress association of Hsp70 with its substrate. In the current study although Hsp70 affinity for peptide was generally reduced in the presence of ATP, the extent to which ATP inhibited this association was dependent on the peptide substrate (Appendix A). Notably, we used hydrolysable ATP whose possible immediate hydrolysis by the respective Hsp70 may account for the marginal reduction in Hsp70 affinity for the substrate.

The ADP and apo state of Hsp70 bind to substrate with comparable affinity. We found this to be the general trend across the various Hsp70-peptide combinations investigated here (Appendix A). With respect to the comparable affinities of the chaperones for the respective peptide and its N enriched form, we focused our attention on the data obtained in the presence of ADP (Figure 2). DnaK bound to its model peptide substrate NRLLTG with higher affinity signals than it exhibited for the respective N-enriched peptides (Figure 2; Appendix A; *p* < 0.01). In fact, the affinity of DnaK for the other two peptides was not enhanced by enriched presence of N residues. On the other hand, the chimera KPf exhibited greater affinity for the peptide NRNNTG than it had for NRLLTG (Figure 2). However, the affinity of KPf for peptides ALLLMYRR and GFRVVLMYRF was not enhanced by enrichment of the peptides with N residues (Figure 2). Interestingly, PfHsp70-1 preferentially bound to all forms of the three peptides that possessed enriched N residues (Figure 2, *p* < 0.05). Overall, these findings suggest that whereas PfHsp70-1 preferred asparagine rich peptides, DnaK binding was inhibited by asparagine enrichment of the peptides. On the other hand, N enrichment only promoted association of KPf with one peptide (NRNNTG). The data suggest that while asparagine rich peptides were preferentially recognized by the SBD of PfHsp70-1, the dynamics of association were regulated in part by the particular ATPase domain attached to the SBD. We previously observed that another plasmodial Hsp70, PfHsp70-x preferentially binds to asparagine rich peptides [25]. In addition, an Hsp70-like member of the Hsp110 subfamily was previously proposed to bind asparagine repeat rich substrates [53]. The current data suggest that SBD of plasmodial Hsp70s are tailored to bind asparagine rich peptide substrates.

As a follow up study, we conducted comparative amino acid sequence analysis and established that asparagine residues occur in AdoMetDC, a possible Hsp70 substrate, at various frequencies: 14.57% in *P. falciparum*, 3.94% in *E. coli* and 3.64% in the human protein, respectively (Figure 3a). Notably, asparagine is by far the most represented amino acid in PfAdoMetDC, and several segments of asparagine residue repeats occur in the protein. The enriched presence of asparagine repeat segments in PfAdoMetDC is interesting, as a previous study we conducted suggested that this protein may be substrate of PfHsp70-1 [24].

### 3.7. SEC Analysis of Recombinant PfAdoMetDC Protein Co-Produced with Supplementary Molecular Chaperones

The recombinant PfAdoMetDC expressed in *E. coli* BL21 Star (DE3) cells endowed with endogenous levels of DnaK or supplementary chaperone sets was purified as previously described (Appendix A; [24]). Based on SEC analysis, monomeric PfAdoMetDC was eluted at the expected retention time at approximately 66 kDa based on MW calibrations used. Interestingly, PfAdoMetDC co-expressed with DnaJ and either DnaK/KPf/PfHsp70-1 eluted at a retention time around approximately 18 min (Figure 3b). On the other hand, recombinant PfAdoMetDC that was co-expressed with DnaK/KPf/PfHsp70-1+DnaJ+GroEL migrated at a slightly reduced pace, suggesting that it assumed a more compact conformation than the protein expressed in the presence of only DnaJ-DnaK/KPf/PfHsp70-1 (Figure 3b). Interestingly, the elution profile of PfAdoMetDC expressed in the absence of supplementary chaperones eluted at nearly the same retention time as the protein co-expressed with supplementary Hsp70 plus GroEL. PfAdoMetDC expressed in the absence of supplementary chaperones is known to aggregate and is hardly biochemically active [24,34], and hence its comparatively long retention time could be due to its mis-folded status Altogether, these findings suggest that supplementary GroEL may have improved folding of PfAdoMetDC towards a more compact conformation that was attained in the presence of DnaJ-DnaK/KPf/PfHsp70-1 chaperone sets.

### 3.8. PfAdoMetDC Co-Produced with PfHsp70-1 and KPf Exhibits Unique Secondary Structural Features

We conducted CD spectroscopy to investigate the secondary structure of PfAdoMetDC co-expressed with the various chaperone sets. The assay was conducted at 22 °C and changes in mean residue ellipticity were monitored at wavelengths 200–240 nm (Figure 3c,d). The assay was repeated in the presence of the SAM, the PfAdoMetDC substrate (Figure 3e,f). Except for the control protein (produced without supplementary chaperones), PfAdoMetDC co-produced with supplementary DnaJ and any of DnaK/KPf/PfHsp70-1 exhibited two distinct negative troughs at 210 and 222 nm, respectively (Figure 3c). The observed spectra are consistent with the predominantly β-sheet fold of PfAdoMetDC [34]. However, the trough for the spectra generated by PfAdoMetDC co-produced with KPf and PfHsp70-1 was deeper and distinct compared to that of PfAdoMetDC co-produced with DnaK. This suggests that PfAdoMetDC co-produced with supplementary DnaK lost some of its ß-sheet fold.

Furthermore, PfAdoMetDC co-expressed with DnaJ+DnaK/KPf/PfHsp70-1 and supplementary GroEL generated a spectrum representing a strong β-sheet fold (Figure 3d). Notably, the inclusion of GroEL to the DnaJ-DnaK chaperone set, resulted in the production of PfAdoMetDC, whose spectrum had a deeper trough than registered by PfAdoMetDC produced in the absence of supplementary GroEL. This suggests that the introduction of GroEL rescued the folding dilemma of PfAdoMetDC co-produced with DnaJ+DnaK (Figure 3d). However, the folding of PfAdoMetDC co-produced with DnaJ+PfHsp70-1/KPf was not modulated by the introduction of supplementary GroEL, suggesting that KPf and PfHsp70-1 were sufficient for folding PfAdoMetDC. Furthermore, the stability of the secondary structural fold of PfAdoMetDC was assessed in the presence of the substrate, SAM (Figure 3e,3f). The data obtained in the presence of SAM, mirrored the data obtained in its absence, highlighting the consistency of the findings. Altogether, these findings suggest that while DnaK confounded PfAdoMetDC folding, the introduction of supplementary GroEL masked the protein folding deficiency of DnaK.

### 3.9. Confirmation of PfAdoMetDC Fold Using ANS Fluorescence-Based Assay

The ANS binds to hydrophobic (nonpolar) surfaces of proteins, through its nonpolar anilino-naphthalene group [39]. For this reason, ANS is used to estimate the levels of exposed hydrophobic clusters of protein. In turn, the exposure of a protein’s hydrophobic residues increases when a protein is mis-folded. In the current study, the folded structures of PfAdoMetDC co-expressed with various combinations of chaperones were analyzed using the ANS fluorescence spectra at wavelengths 425–600 nm (Figure 4a,b). The assay was repeated in the presence SAM (Figure 4c,d). PfAdoMetDC co-expressed with DnaJ + KPf or DnaJ + PfHsp70-1 exhibited the highest fluorescence signal at the same peak. On the other hand, PfAdoMetDC co-produced with DnaJ + DnaK, had a lower peak than that for protein co-produced with DnaJ + KPf or DnaJ + PfHsp70-1 (Figure 4a). PfAdoMetDC expressed in the absence of supplementary chaperones had the lowest fluorescence signal (Figure 4a). The introduction of SAM did not change the fluorescence spectra for PfAdoMetDC co-produced with DnaJ + KPf / PfHsp70-1 / DnaK significantly (Figure 4c). As we observed before, the addition of GroEL to the DnaJ + DnaK led to the generation of PfAdoMetDC, whose folds were similar to those of the protein co-produced with DnaJ + KPf / PfHsp70-1 + GroEL (Figure 4b,d). This further suggests that PfAdoMetDC co-produced with DnaJ + DnaK required supplementary GroEL for complete folding. However, although the introduction of GroEL led to an increase in the fluorescence signal for PfAdoMetDC, it was also associated with a broad spectral peak which tended towards a blue shift (Figure 4b,d). This suggests that PfAdoMetDC co-produced with supplementary DnaJ + DnaK + GroEL was still less folded than that co-produced with supplementary DnaJ + KPf / PfHsp70-1 with/without supplementary GroEL. This suggests that DnaK did not support PfAdoMetDC folding, and while GroEL rescued mis-folded PfAdoMetDC, it could not overcome all the folding deficiencies of PfAdoMetDC induced by co-expressing it with DnaK. Altogether, our findings demonstrated that PfAdoMetDC folding was affected in the following order, depending on the prevailing chaperone conditions: DnaJ + KPf / PfHsp70-1 > DnaJ + DnaK > no supplementary chaperones. However, as previously noted, the introduction of supplementary GroEL to the DnaJ + DnaK combination led to the production of PfAdoMetDC with improved fold.

### 3.10. Analysis of Tertiary Structure of PfAdoMetDC using Intrinsic Tyrosine and Tryptophan Fluorescence

As a follow up study, the intrinsic fluorescence of PfAdoMetDC was analyzed, taking advantage of the presence of a tryptophan residue (38W) located within the α-subunit of the protein. Due to its aromatic character, tryptophan is most often partially or fully buried in the hydrophobic core of protein. Upon disruption of the protein’s tertiary structure, tryptophan becomes solvent exposed, leading to a blue shift in fluorescence [54]. Tryptophan-based fluorescence analysis of the tertiary structure was conducted by monitoring the emission spectra at 310–450 nm after an initial excitation at 295 nm as previously described [33,55]. As previously observed in the absence of supplementary GroEL, the intrinsic fluorescence signal intensity of PfAdoMetDC co-produced under the various chaperone conditions occurred in the following order: DnaJ + KPf > DnaJ + PfHsp70-1 > DnaJ + DnaK > no supplementary chaperones (Figure 4e). This trend remained the same upon repeating the assay in the presence of the substrate, SAM (Figure 4f). PfAdoMetDC produced with supplementary DnaJ + DnaK in the absence of supplementary GroEL registered much less fluorescence intensity than the protein co-produced with either DnaJ + KPf or DnaJ + PfHsp70-1 (Figure 4e,f). This further confirmed that PfAdoMetDC co-produced with supplementary DnaK assumed a unique conformation compared to PfAdoMetDC preparation that was co-produced with either KPf or PfHsp70-1. PfAdoMetDC that was co-produced with supplementary GroEL plus DnaK registered enhanced fluorescence signal, confirming that GroEL assisted PfAdoMetDC to overcome the folding snag it encountered in the presence of DnaK.

The assay was repeated to monitor the combined fluorescence signals of tryptophan and tyrosine as the protein possesses 36 tyrosine residues. The same trend was observed in which PfAdoMetDC produced in the absence of supplementary chaperones exhibited the lowest fluorescence signal, followed by PfAdoMetDC co-produced with DnaK chaperone system (Appendix A). On the other hand, PfAdoMetDC co-expressed with either PfHsp70-1 or KPf exhibited the highest fluorescence signals, further suggesting that it had a unique conformation. However, the introduction of GroEL overcame the functional deficiencies of the DnaK system on PfAdoMetDC folding (Appendix A). Analysis of PfAdoMetDC conformation was repeated in the presence of its substrate, SAM, and the same trend was observed (Appendix A), further confirming the consistency of the observed trend.

## 4. Discussion

PfHsp70-1 plays an important role in the survival and development of *P. falciparum*, the main agent of malaria. There is evidence that notwithstanding their sequence conservation, Hsp70s exhibit specialized functional features across species. To the best of our knowledge this study for the first time demonstrates that PfHsp70-1 preferentially bound to asparagine enriched peptide substrates in vitro. Furthermore, expression of PfHsp70-1 and KPf in *E. coli* improved PfAdoMetDC folding. On the other hand, the enrichment of model Hsp70 peptide substrates with asparagine did not improve their affinity for *E. coli* DnaK. In addition, recombinant PfAdoMetDC folding did not benefit from co-production with supplementary DnaK in *E. coli*. Our findings do not only demonstrate the unique functional features of PfHsp70-1 relative to DnaK but also highlight the role of the SBD of Hsp70 in the interaction of this protein with nucleotide and the Hsp40 co-chaperone. One of the defining features of cytosolic Hsp70s of parasitic origin is the prominent presence of GGMP repeat motifs located in their C-terminal SBD [1,16]. The GGMP motif and other unique residues present in PfHsp70-1 may confer it with unique functional features.

Heterologously expressed forms of DnaK, PfHsp70-1 and their chimeric product, KPf, previously reversed the thermosensitivity of *E. coli dnaK756* cells, whose native DnaK is functionally compromised [27]. The chimera KPf is made up of the NBD of DnaK and shares the same SBD as PfHsp70-1. For this reason, KPf constituted an appropriate tool to explore the unique functional features of PfHsp70-1 relative to E. coli DnaK. Our findings established that KPf exhibits structure-function features that are unique from its parental isotypes. In addition, co-expressing KPf, DnaK, and PfHsp70-1 with recombinant PfAdoMetDC in *E. coli*, led to the production of PfAdoMetDC protein with unique secondary and tertiary conformations. The findings demonstrate that although the SBD regulates the functional specificity of PfHsp70-1, the cooperation of both the SBD and the NBD is important for this process. We established that KPf was nearly as stable to heat stress as PfHsp70-1, and that DnaK in turn was less stable than both KPf and PfHsp70-1 (Figure 1). This suggests that the SBD of PfHsp70-1 which it shared with KPf, conferred stability to both KPf and PfHsp70-1. This is in concert with a previous study [41] which reported that the SBD of PfHsp70-1 is important for the stability of the protein. As such, the SBD of DnaK may account for its comparatively lower stability to heat stress. We previously observed that the C-terminal EEVN residues of PfHsp70-x (*P. falciparum* Hsp70 that is exported to the parasite-infected red blood cell), contributes to the overall stability of the protein [25]. This further suggests a role of the SBD in regulating Hsp70 stability.

The structural features that KPf shared with PfHsp70-1 confirm the important role of the SBD of Hsp70 in modulating Hsp70 function. However, KPf also possesses key structure-function features that set it apart from PfHsp70-1. KPf exhibited much higher affinity for ATP than DnaK (>200-fold) and its affinity for ATP was at least an order of magnitude higher than of PfHsp70-1 (Appendix A, *p* < 0.01). Interestingly, the high affinity for ATP registered by KPf is mirrored by the fact that the chimeric protein was the most conformationally responsive to the presence of ATP (Figure 1c). It is known that ATP and small molecule inhibitors that bind to the NBD modulate the global conformation of Hsp70 through allostery [27,56,57]. Thus, the enhanced conformational changes that ATP induced on KPf may be on account of the unique NBD–SBD interface of this chimeric protein. Interestingly, both KPf and PfHsp70-1 hydrolyzed ATP more effectively than DnaK (Appendix A, *p* < 0.05). It has previously been reported that PfHsp70-1 exhibits higher ATPase activity than Hsp70s of human, bovine and *E. coli* origin [40]. This suggests that the SBD of Hsp70 plays an important role in regulating its ATPase activity.

PfHsp40 is a *P. falciparum* cytosol localized Hsp40 whose structure-function features resemble those of the canonical *E. coli* DnaJ/Hsp40 [17]. PfHsp40 has been shown to stimulate the ATPase activities of cytosol-localized Hsp70s, including PfHsp70-1 and human Hsp70 [17]. Here we demonstrated that PfHsp40 directly bound to all the three Hsp70s and exhibited comparative affinity for KPf and PfHsp70-1. However, its affinity for DnaK was an order of magnitude lower (Appendix A). Furthermore, PfHsp40 stimulated the ATPase activity of KPf more effectively than it modulated the ATPase activities of either PfHsp70-1 or DnaK (Figure 1d; Appendix A, *p* < 0.01). This finding suggests that the NBD of DnaK and the SBD of PfHsp70-1 constituting the domains of KPf, create a structurally unique NBD–SBD interface that promotes efficient hydrolysis of ATP in the presence of PfHsp40. Since the NBD of Hsp70 is highly conserved while its SBD is fairly divergent, the NBD–SBD interface of Hsp70 is regarded as a unique structural entity that regulates its functional specificity [16].

All the proteins were capable of self-association (Table 1). While KPf and PfHsp70-1 exhibited high affinity (nanomolar range in the presence of ATP), DnaK self-association was weaker (micromolar range) under similar conditions. These findings suggest that ATP promoted self-association of the three proteins, and this is in line with a previous independent study which proposed that oligomerization of DnaK is enhanced by ATP [49,58]. Notably, both KPf and PfHsp70-1 exhibited comparably higher affinity for self-association than DnaK. It has been proposed that oligomerization of Hsp70 is mediated by the NBD–SBD interface and the linker segment [46,59]. Since both KPf and PfHsp70-1 share the same SBD and a highly conserved linker motif [60], these two subdomains that the two chaperones share may have accounted for their comparably higher propensity to form oligomers than DnaK.

It has been proposed that roughly 10% of *P. falciparum* proteome is marked by prion-like repeats and that more than 30% of the parasite proteins are characterized by glutamate/asparagine repeat segments [21]. Furthermore, a previous study we conducted demonstrated that the red blood cell exported parasite Hsp70, PfHsp70-x, preferentially binds to peptides enriched with asparagine residues in vitro [25]. For this reason, we explored the substrate binding preferences of PfHsp70-1 relative to KPf and DnaK.

As expected, in the presence of ADP, DnaK displayed high affinity (nanomolar range) for its model substrate, NRLLTG (*p* = 0.05). However, affinity of DnaK for the L–N substitution version of this peptide (NRNNTG) led to a drop in affinity (micromolar range). It is known that DnaK prefers peptides enriched in hydrophobic residues [10]. Similarly, the enrichment of the other two the peptides, ALLLMYRR and GFRVVLMYRF, with asparagine residues, did not enhance DnaK’s affinity for the peptides in the presence of ADP. On the other hand, introduction of asparagine residues led both KPf and PfHsp70-1 to bind the peptide NRNNTG with higher affinity (nanomolar range) than they displayed for NRLLTG (Figure 2; Appendix A, *p* = 0.05). In addition, PfHsp70-1 bound with higher affinity to peptides GFRNNNMYR and ANNNMYRR than it exhibited for GFRVVLMYRF and ALLLMYRR (Appendix A; *p* < 0.05). However, KPf did not exhibit enhanced affinity for the asparagine enriched forms of these two peptides. Overall, the findings demonstrate that the SBD of PfHsp70-1 is biased towards asparagine rich peptides. In addition, the data suggest a role for the ATPase domain in regulating the functional specificity of the SBD.

We previously demonstrated that PfAdoMetDC co-produced with supplementary DnaJ–DnaK was less active than protein co-produced with DnaJ-KPf / PfHsp70-1 [24]. In the current study, we sought to establish whether co-expression of PfAdoMetDC with the various chaperone sets variably modulates its folded status. To this end, we purified recombinant PfAdoMetDC co-produced with the various chaperones we employed here and subjected it to SEC, CD and fluorescence spectrometric analyses. Our SEC analysis suggested that PfAdoMetDC that was co-expressed with DnaK / KPf / PfHsp70-1 + DnaJ + GroEL was more compact than the protein expressed in the presence of only DnaJ-DnaK / KPf / PfHsp70-1 (Figure 3). Thus, supplementary GroEL appeared to marshal PfAdoMetDC folding towards a more compact conformation (Figure 3). Interestingly, the elution profile of PfAdoMetDC expressed in the absence of supplementary chaperones eluted at nearly the same retention time as the protein co-expressed with the respective supplementary Hsp70 plus GroEL. The finding further suggests that the presence of supplementary GroEL modulated PfAdoMetDC to fold in a unique fashion.

The CD spectroscopic analysis confirmed that PfAdoMetDC is characterized by a dominant ß-sheet fold as previously reported [34]. While the co-expression of PfAdoMetDC with DnaJ-KPf/PfHsp70-1 restored the conformation of the protein, co-production of the protein with DnaJ-DnaK led to partial loss of the ß-sheet fold of the protein (Figure 3). This suggests that DnaK did not support PfAdoMetDC folding. However, combining GroEL with DnaJ-DnaK led to restoration of the ß-conformation of PfAdoMetDC (Figure 3). While the introduction of GroEL appears to have modulated the fold of PfAdoMetDC co-produced with DnaK, it did not alter the apparent secondary structural fold of PfAdoMetDC co-produced with KPf/PfHsp70-1. This seems to suggest that the presence of KPf and PfHsp70-1 led PfAdoMetDC to a fully folded status. Subsequent analyses of PfAdoMetDC by ANS and the tryptophan/tyrosine fluorescence data corroborated that DnaK confounded PfAdoMetDC folding and that both KPf and PfHsp70-1 were more effective in facilitating its folding process (Figure 4). Furthermore, based on CD-spectrometry, SEC analysis, and intrinsic fluorescence analyses (Figure 4; Appendix A), GroEL facilitated PfAdoMetDC to overcome the barriers presented by DnaK in its folding pathway. However, ANS-fluorescence analysis revealed that although GroEL enhanced folding of PfAdoMetDC, the slight blue shift of the spectrum generated by the protein suggests that GroEL may not have fully folded PfAdoMetDC compared to the quality of protein co-produced with either KPf or PfHsp70-1.

The folded status of PfAdoMetDC as noted using CD and fluorescence spectrometric assays was validated by repeating the assays in the presence of the PfAdoMetDC substrate, SAM. The inclusion of SAM did not influence the observed conformation of PfAdoMetDC. Overall, the findings demonstrate that the folding fate of PfAdoMetDC was dependent on the supplementary Hsp70 with which it was co-expressed in *E. coli*. In addition, the data demonstrated that GroEL salvaged PfAdoMetDC that battled to fold in the presence of supplementary DnaK.

## 5. Conclusions

By comparing the structure-function features of PfHsp70-1 relative to those of DnaK and the chimera, KPf, we established that the SBD of PfHsp70-1 is endeared to bind asparagine-enriched peptides. This is quite an important attribute, as the *P. falciparum* proteome is fairly represented by glutamate/asparagine-rich candidates [21,22]. Given the propensity of such a proteome to mis-fold, it seems the malaria parasite is endowed with an Hsp70 protein folding machinery that is primed to handle the protein folding demands of the parasite. This is particularly important at elevated temperatures associated with clinical malaria fever manifestations. Furthermore, the current findings are important for our understanding of the functional versatility of Hsp70, in spite of its apparent sequence conservation. In addition, the current findings may aid ongoing therapeutic efforts to identify small molecule inhibitors that selectively target Hsp70.

## Figures and Tables

**Figure 1 biomolecules-10-00856-f001:**
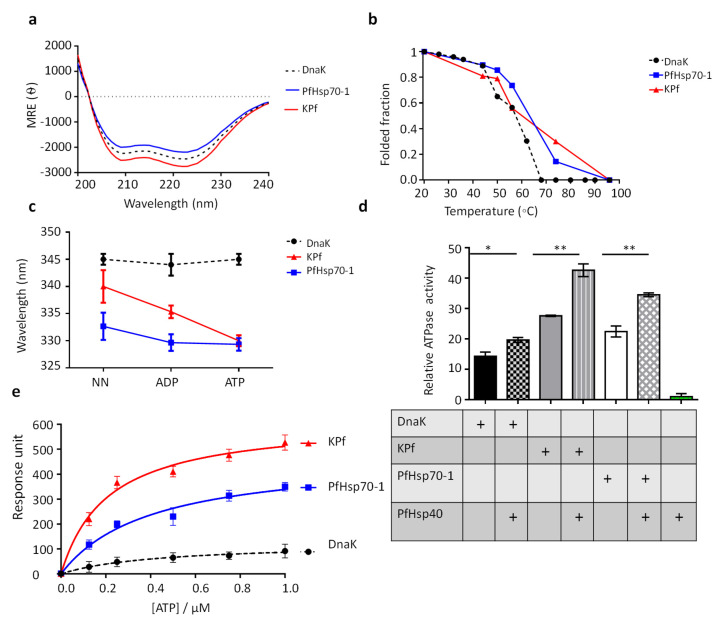
Comparative analysis of heat stability, nucleotide binding and hydrolysis kinetics of the Hsp70s. (**a**) Hsp70 far-UV spectra. The far-UV CD spectra of recombinant DnaK, KPf and PfHsp70-1 proteins were presented as molar residue ellipticity. All the three Hsp70s displayed a predominantly α-helical structure due to the negative troughs observed at 209 and 222 nm. (**b**) Comparative secondary structural fold of DnaK, PfHsp70-1 and KPf exposed to heat stress. The folded fraction of each protein was determined by comparing the spectral readings obtained at any given temperature with readings taken at 19 °C. (**c**) ATP dramatically modulates the conformation of KPf. Fluorescence emission spectra were monitored at 320–450 nm after an initial excitation at 295 nm. The tryptophan fluorescence emission spectra maxima were recorded for DnaK, PfHsp70-1 and KPf proteins in the absence or presence of 5 mM ATP/ADP. The fluorescence maxima for each protein was blue shifted in the presence of either ATP or ADP. (**d**) ATPase activities of PfHsp70, KPf and DnaK. Inorganic phosphate released by ATP in the presence of each Hsp70 was monitored by direct calorimetry at 595 nm wavelength. The observed basal and PfHsp40-stimulated ATPase activities of respective protein were illustrated as bar graphs. The independent activity of PfHsp40 was used as a non-ATPase control. The statistical significance of the relative ATPase activities of the proteins are shown; **p* < 0.05 and ***p* < 0.01 as determined by ANOVA. (**e**) Equilibrium ATP binding kinetics of all three Hsp70s. The reported ATP binding constant for the respective Hsp70 was determined at equilibrium. Standard deviations shown represent three independent assays made using separate protein purification batches.

**Figure 2 biomolecules-10-00856-f002:**
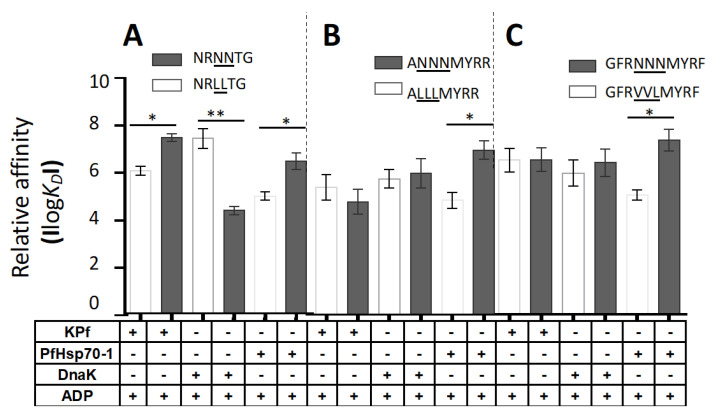
PfHsp70 preferentially bound to asparagine-enriched peptides. The relative affinities of DnaK, KPf and PfHsp70-1 for the various peptides were determined in the presence of 5 mM ADP. Relative affinities for peptides (**A**) NRLLTG versus NRNNTG; (**B**) ALLLMYRR versus ANNNMYRR; (**C**) GFRVVLMYRF versus GFRNNNMYRF are depicted as bar graphs. The error bars shown were generated from three assays conducted using independent Hsp70 protein preparations. Statistical significance was determined by one-way ANOVA (**p < 0.05* and ***p < 0.01*).

**Figure 3 biomolecules-10-00856-f003:**
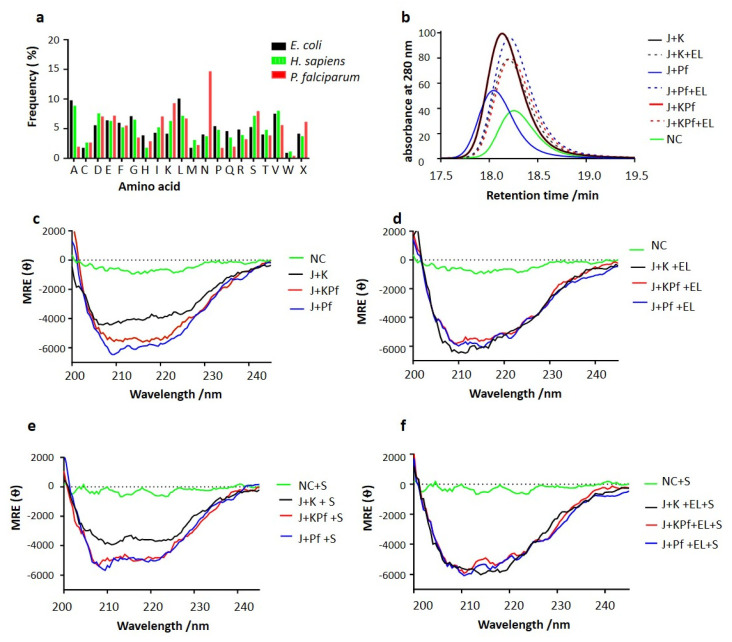
PfAdoMetDC co-produced with PfHsp70-1 and KPf is structurally unique secondary structure from that co-produced with DnaK. Amino acid compositions of AdoMetDC of *H. sapiens, E. coli* and *P. falciparum* origins (**a**). PfAdoMetDC was co-produced either alone (NC) or in the presence of supplementary chaperone sets: DnaK + DnaJ (J + K); DnaK + DnaJ + GroEL (J + K + EL); PfHsp70-1+DnaJ (J + Pf); PfHsp70 + DnaJ + GroEL (J + Pf + EL); KPf + DnaJ (J + KPf); and KPf + DnaJ + GroEL (J + KPf + EL), respectively. The protein was then subjected to SEC-FPLC analysis (**b**); and far-UV CD spectrometric assays (**c**–**f**). CD spectrometric analysis was conducted either in the absence of substrate or in the presence of SAM (‘S’), the PfAdoMetDC substrate. The CD spectrum of PfAdoMetDC was presented as molar residue ellipticity (MRE; θ).

**Figure 4 biomolecules-10-00856-f004:**
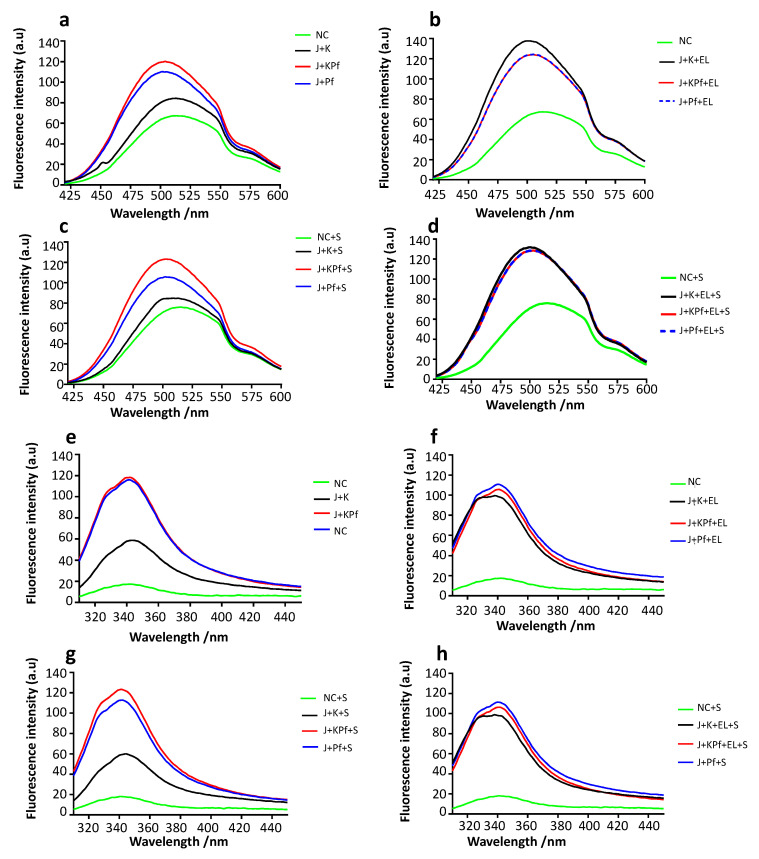
Co-production of PfAdoMetDC with PfHsp70-1 and KPf facilitates the folding process. PfAdoMetDC co-expressed with the various chaperone sets was subjected to either ANS extrinsic fluorescence analysis (**a**–**d**) or intrinsic fluorescence analysis (**e**–**g**). PfAdoMetDC produced in the absence of supplementary chaperones (NC) or presence of various heterologous chaperones combinations: DnaK + DnaJ (J + K); DnaK + DnaJ + GroEL (J + K + EL); PfHsp70 + DnaJ (J + Pf); PfHsp70 + DnaJ + GroEL (J + Pf + EL); KPf + DnaJ (J + KPf); KPf + DnaJ + GroEL (J + KPf + EL). The fluorescence analysis was conducted in the absence or presence of the SAM (S), the PfAdometDC substrate.

**Table 1 biomolecules-10-00856-t001:** Kinetics for self-association of DnaK, KPf and PfHsp70-1.

Ligand	Analyte	*K_a_* (Ms^−1^)	*K_d_* (1^−1^)	*K_D_* (M)	*X* ^2^
DnaK	DnaK + ATP	1.38 (±0.08) × 10^2^	3.83 (±0.33) × 10^−4^	4.13 (±0.30) × 10^−6^*	5.15
	DnaK	1.27 (±0.07) × 10^3^	6.51 (±0.01) × 10^−2^	4.67 (±0.27) × 10^−5^	4.21
	DnaK + ADP	1.22 (±0.13) × 10^2^	6.49 (±0.09) × 10^−3^	4.60 (±0.09) × 10^−5^	5.41
KPf	KPf + ATP	1.54 (±0.04) × 10^2^	5.44 (±0.04) × 10^−3^	3.23 (±0.03) × 10^−7^*	2.30
	KPf	1.32 (±0.02) × 10^2^	4.17 (±0.17) × 10^−5^	5.38 (±0.08) × 10^−6^	3.17
	KPf + ADP	1.42 (±0.02) × 10^2^	5.23 (±0.03) × 10^−3^	4.65 (±1.5) × 10^−5^	2.37
PfHsp70-1	PfHsp70-1 + ATP	2.14 (±0.04) × 10^4^	1.13 (±1.2) × 10^−2^	5.28 (±0.08) × 10^−7^*	4.42
	PfHsp70-1	8.51 (±1.20) × 10^3^	2.04 (±0.11) × 10^−3^	2.39 (±0.09) × 10^−6^	1.20
	PfHsp70-1 + ADP	1.00 (±0.17) × 10^2^	1.71 (±0.07) × 10^−4^	1.71 (±0.10) × 10^−6^	4.45

The table shows the binding kinetics parameters of self-association of the three Hsp70s. The ligand fraction of each protein was the immobilized protein on the GLC chip surface, and the analyte fraction was injected at a flow rate of 50 μL/min. Self-association of the three proteins was investigated either in the absence or presence of ATP/ADP. Significant variation in K_D_ values of data obtained in the presence of ATP was based on one-way ANOVA test (**p* < 0.05).

**Table 2 biomolecules-10-00856-t002:** Kinetics for the interaction of DnaK/KPf/PfHsp70-1 with PfHsp40.

Ligand	Analyte	*K_a_* (Ms^−1^)	*K_d_* (1^−1^)	*K_D_* (M)	*X* ^2^
PfHsp70-1	PfHsp40 +ATP	1.81 (±0.01) × 10^2^	1.32 (±0.02) × 10^−^^4^	2.08 (±0.80) × 10^−^^7^ **	2.12
	PfHsp40 +ADP	1.53 (±0.03) × 10^3^	8.66 (±0.06) × 10^−^^6^	9.98 (±0.18) × 10^−^^6^	3.08
	PfHsp40	1.33 (±0.03) × 10^3^	9.81(±0.10) × 10^−^^6^	1.88 (±0.08) × 10^−^^5^	7.80
KPf	PfHsp40 + ATP	1.72 (±0.02) × 10^2^	7.44 (±0.04) × 10^−^^3^	7.23 (±0.30) × 10^−^^7^**	1.75
	PfHsp40 + ADP	1.36 (±0.26) × 10^2^	5.43 (±0.03) × 10^−^^4^	2.62 (±0.02) × 10^−^^6^	2.15
	PfHsp40	1.24 (±0.04) × 10^4^	6.54 (±0.04) × 10^−^^5^	2.53 (±0.03) × 10^−^^5^	2.13
DnaK	PfHsp40 + ATP	1.22 (±0.02) × 10^2^	7.44 (±0.04) × 10^−^^4^	7.23 (±0.03) × 10^−^^6^*	1.66
	PfHsp40 + ADP	1.16 (±0.06) × 10^2^	5.43 (±0.32) × 10^−^^3^	2.62 (±0.02) × 10^−^^5^	1.57
	PfHsp40	1.12 (±0.02) × 10^2^	7.44 (±0.04) × 10^−^^3^	7.23 (±0.03) × 10^−^^5^	2.07

The data were analyzed through global fitting of the sensorgrams obtained. The standard deviations about the means were obtained after three repeat experimental runs. Statistical significance of the data obtained in the presence of ATP was established by one-way ANOVA test (**p* < 0.05) and (***p* < 0.01).

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
