# Peer review of "Comparative Characterization of Plasmodium falciparum Hsp70-1 Relative to E. coli DnaK Reveals the Functional Specificity of the Parasite Chaperone"

_biomolecules, 2020, doi:10.3390/biom10060856_

Round 1

Reviewer 1 Report

Lebepe and Colleagues investigate the impact of different forms of Hsp70s on the ATPase, ATP binding and folding status of Plasmodium protein PfAdoMetDC with help of cochaperone protein PfHsp40. The comparative structure-function studies reveal that the SBD of PfHsp70-1 is important for its specificity in the proper folding of Plasmodium protein in E coli.

Overall, this is a carefully designed study that provides much novel insight into the specific function of malaria chaperone, many aspects of which will also be important for anti-malaria drug development. The manuscript is well-written, and I shall gladly recommend it be published in Biomolecules following some revision.

Major comments

1 The authors imply that PfAdoMetDC is an asparagine-rich protein, therefore the strong affinity of PfHsp70-1 and KPf to the peptides with asparagine repeat is the basis for proper folding of PfADoMetDC in E coli, as opposed to DnaK. But they fail to provide evidence that PfAdoMetDC is rich in asparagine. A thorough analysis of PfAdoMetDC amino acids sequence should be provided in the main text to make a smooth transition from asparagine repeat binding by PfHSP70-1 to proper folding of PfAdoMetDC. ­­

2 The role of GroEL in DnaK dependent folding of PfAdoMetDC was not summarized in Abstract section.

3 The authors inappropriately used hyphen (-) or underscore (_) symbols for Plasmodium falciparum ORFs. It appears several times in text for Hsp40 coding gene PF3D7_1437900 (should be underscore) to be mis-written by the authors as PF3D7-1437900 (hyphen symbol)

4 For clarity and readability, the color and shape scheme for denoting each Hsp70 chaperone should remain consistent throughout the manuscript. For example, if the authors decide to use black triangle for PfHsp70-1 in one figure, they should not use black square or red square for PfHsp70-1 in other figures. In addition, solid line or broken line scheme should also be consistent throughout the manuscript.

5 Is the relative ATPase activity in Fig 1d plotted using the Vmax data in Table S1 or is it proportional to Vmax in TableS1. If yes, the error bars in Fig1d do not match with standard deviation in Table S1.

6 In Table S1, it appears that PfHsp40 confers lower Km to PfHsp70 vs PfHSp70 alone. This effects on Km of PfHsp70 should be discussed.

7 In terms of Molar residue ellipticity (MRE), two different scales were used in Fig 1a and Fig 3b, c, d, e. The authors may want to make the y-axis consistent.

8 I cannot find J+KPf in Fig 3a.

Minor comments

1 The first time appearance of the abbreviation “SEC” should follow the full description “size exclusion chromatography”

2 Table 2 The labels of kinetic parameters for each column are missing.

3 The legend for Fig 1d was mis-labeled as Fig 1c, there are two (C) in Fig1 legend.

4 Page 10 line 417, where the authors refer to Fig2, Fig S3 for the statement “a particular substrate in the apo state or in the presence of ATP/ADP, the figures relevant should be Fig2 and Fig S2, not Fig S3.

5 Page 15 line 571 Fig S4 was referred as Fig S3

6 I was confused by Fig S3, what antibodies were used for Western blot, each Hsp70 specific antibody or anti-His antibody since no antibody information was provided in the “material and methods” for Hsp70 specific antibodies.

Author Response

Author Responses

Reviewer #1

Lebepe and Colleagues investigate the impact of different forms of Hsp70s on the ATPase, ATP binding and folding status of Plasmodium protein PfAdoMetDC with help of cochaperone protein PfHsp40. The comparative structure-function studies reveal that the SBD of PfHsp70-1 is important for its specificity in the proper folding of Plasmodium protein in E coli.

Overall, this is a carefully designed study that provides much novel insight into the specific function of malaria chaperone, many aspects of which will also be important for anti-malaria drug development. The manuscript is well-written, and I shall gladly recommend it be published in Biomolecules following some revision.

Major comments

 1 The authors imply that PfAdoMetDC is an asparagine-rich protein, therefore the strong affinity of PfHsp70-1 and KPf to the peptides with asparagine repeat is the basis for proper folding of PfADoMetDC in E coli, as opposed to DnaK. But they fail to provide evidence that PfAdoMetDC is rich in asparagine. A thorough analysis of PfAdoMetDC amino acids sequence should be provided in the main text to make a smooth transition from asparagine repeat binding by PfHSP70-1 to proper folding of PfAdoMetDC. ­­

Author’s response: We thank the reviewer for this suggestion. The multiple sequence alignment was previously published (Birkholtz, L.M., Wrenger, C., Joubert, F., Wells, G.A., Walter, R.D. and Louw, A.I., 2004. Parasite-specific inserts in the bifunctional S-adenosylmethionine decarboxylase/ornithine decarboxylase of Plasmodium falciparum modulate catalytic activities and domain interactions. Biochemical Journal, 377(2), pp.439-448). For this reason, we chose to conduct amino acid residue frequency in AdoMetDC homologues of E. coli, human and P. falciparum. Interestingly, N residues are evidently enriched in the latter (see revised Figure 3a).

2 The role of GroEL in DnaK dependent folding of PfAdoMetDC was not summarized in Abstract section.

Author’s response: The following statement has been inserted in the abstract. “However, a combination of supplementary GroEL plus DnaK improved folding of PfAdoMetDC.” 

3 The authors inappropriately used hyphen (-) or underscore (_) symbols for Plasmodium falciparum ORFs. It appears several times in text for Hsp40 coding gene PF3D7_1437900 (should be underscore) to be mis-written by the authors as PF3D7-1437900 (hyphen symbol)

Author’s response: This has been corrected.

4 For clarity and readability, the color and shape scheme for denoting each Hsp70 chaperone should remain consistent throughout the manuscript. For example, if the authors decide to use black triangle for PfHsp70-1 in one figure, they should not use black square or red square for PfHsp70-1 in other figures. In addition, solid line or broken line scheme should also be consistent throughout the manuscript.

Author’s response: This has been done through-out the whole MS with KPf in Red, DnaK in Black and PfHsp70-1 in blue.

5 Is the relative ATPase activity in Fig 1d plotted using the Vmax data in Table S1 or is it proportional to Vmax in TableS1. If yes, the error bars in Fig1d do not match with standard deviation in Table S1.

Author’s response: We acknowledge this error and have since redrawn the graph using  the Vmax values presented in Table S1.

6 In Table S1, it appears that PfHsp40 confers lower Km to PfHsp70 vs PfHSp70 alone. This effects on Km of PfHsp70 should be discussed.

Author’s response: The following statement was inserted “As a bona fide co-chaperone of PfHsp70-1, PfHsp40 modulated the ATPase activity of PfHsp70-1 registering more than ten-fold reduction in Km value compared to the basal ATPase activity of the chaperone [Table S1; 17].”

7 In terms of Molar residue ellipticity (MRE), two different scales were used in Fig 1a and Fig 3b, c, d, e. The authors may want to make the y-axis consistent.

Author’s response: It has been corrected to be consistent into MRE (q) throughout the MS.

8 I cannot find J+KPf in Fig 3a.

Author’s response: We appreciate the reviewer’s comments and we have changed the colour codes to make this graph more visible as it was previously masked under overlapping line representing J+K.

 Minor comments

  1. The first time appearance of the abbreviation “SEC” should follow the full description “size exclusion chromatography”

Author’s response: We have defined SEC acronym on first mention.

  1. Table 2 The labels of kinetic parameters for each column are missing.

Author’s response: We have added the subheading to Table 2 as follows:

Ligand

Analyte

Ka (Ms-1)

Kd (1-1)

KD (M)

X2

  1. The legend for Fig 1d was mis-labeled as Fig 1c, there are two (C) in Fig1 legend.

Author’s response: We have corrected the panel as figure 1d.

  1. Page 10 line 417, where the authors refer to Fig2, Fig S3 for the statement “a particular substrate in the apo state or in the presence of ATP/ADP, the figures relevant should be Fig2 and Fig S2, not Fig S3.

Author’s response: We have aligned the numbering to go with the new supplementary figure numbering.

  1. Page 15 line 571 Fig S4 was referred as Fig S3.

Author’s response: We have corrected the reference to Fig S4 as we have added another supplementary Figure as S1 for homology model data.

  1. I was confused by Fig S3, what antibodies were used for Western blot, each Hsp70 specific antibody or anti-His antibody since no antibody information was provided in the “material and methods” for Hsp70 specific antibodies.

Author’s response: We have highlighted the antibodies used in the materials section as follows: “The following antibodies were used; rabbit raised polyclonal anti-PfHsp70-1 (32), mouse raised monoclonal anti-DnaK (27), mouse raised monoclonal anti-Strep-tag II (24; Novagen, Wisconsin, USA), goat raised anti-rabbit horseradish peroxidase (HRP) conjugated (Sigma-Aldrich (USA), goat raised anti-mouse HRP conjugated (Sigma-Aldrich (USA), monoclonal α-His-HRP conjugated (ThermoScientific, USA”.

Reviewer 2 Report

This paper is potentially interesting. It demonstrates that the plasmodial chaperone PfHsp70-1 preferentially bound to asparagine-enriched peptide substrates in vitro. However, the seminal study Plasmodium falciparum heat shock protein 110 stabilizes the asparagine repeat-rich parasite proteome during malarial fevers was not even mentioned. In this seminal study, it was shown that hsp110, probably together with hsp70 protein, stabilizes asparagine repeat regions.

Furthermore, the paper attempts to characterize features of PfHsp70-1 relative to DnaK and a chimeric protein, KPf. The paper indicates that it is likely that the structures of these proteins are similar. However, homology modeling was not applied. As a result, the paper does not provide any explanations for substrate specificity; instead, it demonstrates that hsp70 proteins have similar three-dimensional structures. This is rather obvious. In my opinion, the paper should be based on homology modeling.

Author Response

Reviewer feedback

Reviewer #2

This paper is potentially interesting. It demonstrates that the plasmodial chaperone PfHsp70-1 preferentially bound to asparagine-enriched peptide substrates in vitro. However, the seminal study Plasmodium falciparum heat shock protein 110 stabilizes the asparagine repeat-rich parasite proteome during malarial fevers was not even mentioned. In this seminal study, it was shown that hsp110, probably together with hsp70 protein, stabilizes asparagine repeat regions.

Author’s response: This study was focusing on canonical Hsp70 family members and we didn’t want to divert the focus to include the Hsp110 family as it was beyond the scope of this study. However, we have included the reference “54. Muralidharan, V.; Oksman, A.; Pal, P.; Lindquist, S; & Goldberg, D.E. Plasmodium falciparum heat shock protein 110 stabilizes the asparagine repeat-rich parasite proteome during malarial fevers. Nat Comm, 2012; 3, 1-10”.

Furthermore, the paper attempts to characterize features of PfHsp70-1 relative to DnaK and a chimeric protein, KPf. The paper indicates that it is likely that the structures of these proteins are similar. However, homology modeling was not applied. As a result, the paper does not provide any explanations for substrate specificity; instead, it demonstrates that hsp70 proteins have similar three-dimensional structures. This is rather obvious. In my opinion, the paper should be based on homology modeling.

Author’s response: We have conducted the homology modelling for the three proteins and due to the varying conformational states in solved templated in nucleotide bound states. We opted to show the subdomain structures after subjecting the models to structural comparison using the matchmaking tool on chimera as supplementary (Fig S1).